# Exploring the Gamification Affordances in Online Shopping with the Heterogeneity Examination through REBUS-PLS

Xiao-Yu Xu [1,*], Syed Muhammad Usman Tayyab [2] , Qing-Dan Jia [1] and Kuang Wu [1]

1   School of Economics and Finance, Xi'an Jiaotong University, Xi'an 710061, China
2   Desautels Faculty of Management, McGill University, Montreal, QC H3A 1G5, Canada
*   Correspondence: xuxiaoyu@xjtu.edu.cn

**Abstract:** This study investigates, from the perspective of affordance theory, how the implementation of gamification features and mechanisms in online-shopping platforms enable consumers to enjoy immersive shopping experiences and make subsequent shopping decisions. Importantly, the technique of REBUS-PLS is applied to unveil the nature of heterogeneity in perceived affordances and ensure the robustness of structural-model results. The research model is tested using cross-sectional data. Our results not only confirm the effects of different types of gamification affordances on immersive experience and subsequent behavior but also reveal the existence of different consumer groups within the overall sample with respect to their behavior patterns. Apart from social connectiveness, rewardability, playfulness, and novelty all exert significant effects on the immersive experience. In addition, this study identified three distinct groups, namely, "no novelty" users, "no playfulness" users, and "no connective" users.

**Keywords:** gamification; online shopping; affordance theory; heterogeneity





## 1. Introduction

It has been a new trend to embed game mechanisms into nongame contexts to encourage users' behavioral changes [1,2]. The marketing report predicts that the gamification market will maintain a compound annual growth rate of 44.06% from 2018 to 2023, thus reaching a global market value of USD 19.4 billion in 2023 [3]. It is believed that gamification can have a great impact in different fields, such as sustaining customers in retail, assisting learning behavior change in education [4], encouraging healthy behavior [5], etc. Significant improvements generated from gamification are observed, such as increased numbers of 500% and 66% in customer product reviews and website visits, respectively, on Samsung nation's website [6].

In a gamification-embedded online-shopping environment, consumers are empowered to immerse themselves in gaming activities and obtain unique benefits that are not possible with traditional online shopping, such as obtaining monetary rewards and getting to know the products while playing games [7,8]. For example, the e-commerce giant Alibaba designed "Ali-Farm," which allows users to fertilize virtual fruit trees to win real fruits for free. Consumers can participate actively in multiple tasks to win more virtual fertilizers, such as by viewing or buying recommended products, inviting friends to complete the gaming tasks together, etc. Thus, consumers often spend time and effort understanding the action possibilities during their interaction with shopping game systems [9]. However, studies on gamification in online shopping are still in their embryonic stage. Most researchers believe gamification is applied in loyalty or rewards programs, and self-determined theory and technology-acceptance theory is useful to explain why gamification works [1]. Several studies hold an assumption that users respond passively to technology (gamification elements) or environment stimuli [10]. In addition, since the phenomenon is emerging, many studies have applied self-developed models and laboratory experiments to explore the effects

of specific elements on users' psychological states and behaviors [10–12]. These studies offer useful knowledge about gamification; however, these approaches provide insufficient justification regarding the activities in which customers can participate actively in gamification-embedded online shopping.

The concept of affordance is useful to investigate our research focus [13], since the affordance lens treats users as active participants who explore the possibilities or actions while interacting with an artifact (e.g., gamification). Importantly, affordances are context-laden, meaning that the impact of multiple affordances on users' attitudes, reactions, and behaviors varies in different scenarios. In other words, the affordances identified in other contexts are not completely applicable in gamification-embedded online-shopping environments (e.g., [14]). Thus, this research gap calls for the contextualization of affordance theory in gamified online shopping [15]. Such investigation can offer insights into the literature on gamification and e-commerce. In addition, the exploration of the affordance lens should be further expanded. Since the affordances identified in different studies may vary, the differentiated affordances offer few opportunities for results comparison and accumulation of knowledge. Thus, it raises the need for a unifying categorization that can encompass a variety of different affordances in different contexts.

To better contextualize affordance theory in our research context, a proper research approach should be carefully designed based on an in-depth understanding of the concept of affordance. It is possible for different objects to be utilized by people in a similar manner (or for similar objects to be used in a different manner) based on perceptions of the affordances in specific contexts [16]. In other words, people may show heterogeneous behavior patterns based on their perception of a specific affordance. More specifically, online-shopping consumers do not draw affordances from the gamification phenomenon in similar ways. Thus, it can be safely argued that the affordance perspective naturally lends itself to the notion of latent classes of users exhibiting heterogeneous behaviors. However, most studies that use PLS-SEM to test their affordance models simply overlook the potential effects of the presence of different segments of users (latent classes) on the proposed structural model. A few studies that consider heterogeneity under affordance-related studies simply opt to control for it instead [17]. This negligent approach clearly undermines the basic tenets of the affordance perspective, which draws from the relational link between users and technology.

From a methodological perspective, researchers employing the structural-equation-modeling technique (SEM) often assume that data originate from a homogenous user population. However, there is strong evidence in the literature that opposes this homogeneity assumption [18]. The latest partial-least-squares structural-equation-modeling (PLS-SEM) guidelines make an explicit case for considering unobserved heterogeneity [19,20]. The response-based procedure for detecting unit segments–partial least squares (REBUS-PLS) can detect the distinct segments of users present in the data based on the hypothesized correlations between the constructs outlined in the proposed path model, which are not known a priori. Since many prior studies employing SEM do not address the presence of latent classes (unobserved heterogeneity) they thus undermine the validity of any subsequent empirical analysis. Hence, the application of REBUS-PLS can address the research gaps remaining in the literature.

To fill the aforementioned research gaps, we employed affordance theory to investigate how gamification affordances influence purchase intentions in gamified shopping settings. In addition, this study endeavors to offer a unifying categorization that can encompass the identified affordances. To understand the influence mechanism of gamification affordances on customer purchasing decisions, immersion is employed as the key element between affordances and subsequent behavior in gamified shopping contexts [21–23]. Moreover, we propose to identify the gamification affordance classification and the representative affordance for each category in an e-commerce setting. To address the heterogeneous behavior patterns in the gamified shopping context, REBUS-PLS is proposed to expound user behavior under this heterogeneous scenario. In the REBUS-PLS approach, the latent

classes of users in terms of the proposed inter-constructs relationship are initially identified (in light of measurements and the structural model). Then, a moderation analysis is conducted based on the identified latent classes/segments.

Therefore, this study contributes to the literature on gamification and affordance in two ways. First, we interpreted consumers' decision mechanisms in gamification embedded in e-commerce from the perspective of affordance theory, extending the contextualization of affordance theory and providing useful knowledge to the literature on gamification and online consumers' behaviors. Second, we analyzed the effects of unobserved heterogeneity and potential moderators in an affordance–behavior relationship under gamified online-shopping scenarios by combining PLS-SEM analysis with REBUS. The integration of REBUS not only addresses the basic assumption of affordance but also produces methodological implications.

## 2. Theoretical Background and Literature Review

### 2.1. Gamification

Gamification refers to the employment of game mechanisms (or elements) to provide affordances for enhancing gameful experiences in non-game contexts and influence user behaviors [12,24]. Gamification has been employed across disciplines such as education [25], knowledge management [26], branding [27], social media [28], and online fitness [29–31]. Researchers believe gamification can produce a variety of benefits, such as enhancing user interaction, offering novel stimuli, and improving user satisfaction, engagement, and loyalty [32]. Gamification marketing activities can impact brand love [2,33,34]. Moreover, users are often attracted to a variety of gaming activities, such as completing gaming tasks, obtaining monetary rewards, and indulging in a new form of social interaction [9,35]). Researchers highlight the employment of game elements to craft gameful experience such as satisfaction, enjoyment, and immersion, since gameful design is effective at engaging and exciting individuals in a playful scenario [7,8].

Researchers suggested that gamification practice can increase motivation for specific actions or behaviors [8]. The inclusion of gamification in online shopping is acknowledged as one of the most prominent approaches for offering more exciting and gameful activities, creating an enjoyable experience, and promoting decision-making processes in the e-commerce context [1,7]. For example, consumers are allowed to gain virtual rewards (i.e., point collection) and upgrade a badge or leaderboard by playing shopping games [12,36]. Shi et al. [7] believe that a variety of possibilities and behaviors enabled by gamification (e.g., competition and self-expression) can facilitate tourists' perceptions of diverse values in online travel-agency platforms. Several researchers investigated how the actions of earning monetary rewards in gamified online shopping can enhance customer loyalty [10]. Therefore, the action possibilities enhanced by gamification in e-commerce are significant for this phenomenon.

Prior studies have explored gamification in online marketing from a variety of perspectives. Hsu and Chen [33] focused on the gamification experience and its effect on customer value, which may crease brand love. Tobon et al. [1] conducted a literature review and suggested that most studies believe gamification is applied in loyalty or rewards programs, and that self-determination theory and technology-acceptance theory are useful to explain why gamification works. Several researchers explored how a loyalty program with or without gamification may impact consumer loyalty differently [10,37]. Zhang et al. [12] believe that gamification offers different mechanisms, such as economic and achievement mechanisms, to influence consumers' impulse buying in "double eleven" annual shopping events. García-Jurado et al. [36] suggested that gamification has direct effects on different types of engagement in e-commerce. Shi et al. [7] investigated the relationships between four specific gamification affordances and customer value on online travel-agency platforms. Xu et al. [11] applied cognitive-evaluation theory to investigate the extrinsic and intrinsic motivation of participating in gamification that can positively enhance enjoyment in online-shopping platforms. These studies offer rich results from diverse perspectives,

and only a handful of studies have investigated how gamification activities can empower consumers' action and behavior possibilities that could enhance the immersive experience. Therefore, this investigation focuses on how the empowerment of gamification activities may reveal an understanding of the phenomenon of interest from a different perspective.

### 2.2. The Affordance Lens

### 2.2.1. Gamification Affordance in E-Commerce

Affordance is the possibilities for action that the environment offers to individuals [38]. Affordance denotes the possible interactions between features of objects (e.g., an artifact or IT) and the capabilities of actors [39]. Thus, affordance theory adopts the relational perspective that possibilities for action emerge as a result of interaction between the technological artifacts and the users of the technology within a particular context [40]. Perceived gamification affordances in online shopping are defined as consumers' perceptions of action possibilities enabled by the gamification mechanism in an e-commerce environment, such as the possibility of obtaining novelty, obtaining playfulness and rewards, and socializing during the shopping process [15]. As a novel technology, gamification has the capability of offering unique affordances to its users, developing bonded relationships with consumers, and promoting consumers' immersion in the gamified online-shopping experience. Hence, affordance theory provides us with a unique and apposite perspective to interpret how users' behaviors are evoked by the gamification affordances embedded in online shopping.

Affordances are context-laden; thus, the affordances in a specific setting may be different from the affordances in other contexts to a great extent. For example, IT affordances such as visibility, meta-voicing, and guidance shopping were identified as social-commerce affordances [9], whereas autonomy support and visibility of achievement were investigated as gamification affordances in environmental conservation [41]. Therefore, a study focusing on a specific research context, such as gamification embedded in online shopping, can help with building precise and detailed knowledge contributing to the literature on gamification and e-commerce simultaneously.

To identify the factors capturing gamification affordances, this study employed a two-step process based on a solid literature review and a comprehensive understanding of the phenomenon (please refer to Table S1). The first step was to classify the affordance factors based on a comprehensive literature review related to gamification. The second step was to identify the representative construct for each classification to reflect the unique and contextualized features in our research context. This step is important since a study focusing on specific constructs reflecting the research context can contribute precise and detailed knowledge to the literature [42,43]. Thus, we endeavored to employ the most representative construct to reflect the general type of affordances based on both the feature of the context and the literature review.

Based on the two-step approach, the literature review unveiled that though the specific affordances identified in different studies vary largely based on their context, they can be classified into three categories applying a general taxonomy, namely, utilitarian, connective, and hedonic affordances. The categories were identified to study personal ICT usage by Scheepers & Middleton [44] and verified in several contexts such as social commerce [15]. Scholars suggest that individuals adopt multiple types of affordances to facilitate their experience of technology usage; these affordances can provide utilitarian, social, and hedonic satisfaction. This raises the need for a unifying categorization that can encompass a variety of different affordances in a variety of contexts. These three-dimensional classifications referred to as utilitarian, social, and hedonic affordances can be applied across multiple contexts.

In order to examine how these three categories may be applied in the field of interest, this study conducted a literature review and demonstrated that the classification of the specific affordances complies with these three categories. Using the keywords "gamification," "affordance," and "e-commerce" or their synonyms, we collected the academic articles

from the past five years from the Web of Science bibliographic database, which represents the largest database of research literature. Though we could barely identify the literature investigating gamification affordances in e-commerce, we reviewed the academic articles that either addressed gamification affordance or affordance in e-commerce. The literature review revealed that the specific affordances identified in prior literature could be well classified into these three categories, as shown in Table S1. For example, reward [23,45,46], visibility, and meta-voicing [9,47,48] were classified as a utilitarian affordance. Social affordances may include feedback [45], interactivity [41], and self-expression [7,23]. Therefore, we studied gamification affordances in e-commerce under three categories, namely, utilitarian affordances, hedonic affordances, and social affordances. Since affordances are context-laden, the affordances identified to represent these three categories should be closely related to the attributes of gamification embedded in e-commerce.

Utilitarian affordance refers to the perceived and actual properties of an ICT that allows the individual to perform an action to obtain utilitarian benefits [44]. For example, utilitarian affordance in the consumption context may refer to the action possibilities that a consumer perceives that a shopping environment allows in order to facilitate their shopping decisions [15,49]. It is most common and natural for consumers to buy a product with less money or a discount. In other words, monetary benefit is a common factor to facilitate shopping decisions. Thus, shopping games carefully design the mechanism of providing monetary incentives to consumers, especially when the gaming tasks are embedded with advertisements and products. For example, consumers need to feed virtual pets three times using the pet food obtained by browsing the recommended products. Monetary rewards, coupons, discounts, and even free products are offered to consumers who finish the gaming task. Therefore, monetary rewardability can present utilitarian affordance in shopping games, which provide the action possibilities that a consumer perceives to obtain monetary benefits. In addition, monetary rewardability has been identified as one of the most significant gamification affordances in prior studies [23,45,46], since the affordance of monetary rewardability is a strong incentive that can arouse the participant's enthusiasm for completing gaming tasks.

Hedonic affordance refers to the perceived properties of an ICT that allows individuals to use it as a means of entertainment or leisure [44]. In a shopping scenario, perceived hedonic affordance is understood as action possibilities that lead a consumer to perceive a shopping environment as entertainment or leisure [15]. In the context of gamification embedded in online shopping, consumers are enabled to obtain two typical hedonic affordances, including the hedonic affordance obtained from playing shopping games and the hedonic affordance derived from enjoying the novel shopping activities facilitated by games. First, consumers may perceive playing shopping games as a means to entertain themselves, relax, and spend time. The affordance of playfulness is defined as using gamified shopping as a means of entertainment/leisure in shopping games [50]. Second, novelty refers to the action possibilities in which the environment facilitating shopping games allows consumers to seek new, unique, and different experiences [51]. In particular, shopping games are used as a means to arouse consumers' curiosity and inspire the consumers to immerse in new experiences, such as integrating gamification elements into the shopping process and exploring shopping activities empowered by new gamified stimuli [52–54].

Social affordances mean the personal use of ICTs as a medium to connect to other individuals, systems, or technologies [44]. Perceived social affordances in the shopping context refer to how a consumer perceives that a shopping environment gives them the opportunity to interact with other people [15]. However, in a gamification-embedded shopping environment, the connections among shopping-game consumers are often gamified, and the content of interactions is related to achieving shopping-game goals. For example, a consumer would invite his friend to join him in watering a virtual fruit tree. Individuals are enabled to connect with each other in this gamified context, such as discussing the most efficient strategy to obtain more virtual water for the virtual tree and reach an agreement to

help each other complete the gaming task. When the virtual fruit ripens in the gamified virtual context, the real fruit is delivered for free to both the consumer and his friend. Thus, shopping games can be used by consumers as a medium to connect with other individuals [55]. The connectiveness in this study is defined as how a consumer perceives that a gamification-embedded shopping environment gives them the opportunity to interact with their friends, families, and peers.

### 2.2.2. Affordance and Unobserved Heterogeneity

The notion of unobserved heterogeneity is not new in behavioral science, but still, most of the empirical behavioral and social research assumes homogeneity of the data. In other words, the whole dataset is supposed to be drawn from a homogenous population and well represented by a single global model. This assumption may, however, often turn out to be false. Considering the complexity of behavioral and social phenomena, it is quite likely for unobserved heterogeneity to be present in data used for such studies. In many cases, it is reasonable to expect the existence of different segments of units in the data to result in heterogeneous behavior, and treating all these segments as belonging to a homogenous dataset may lead to biased results in terms of both model parameters and validation indexes [20,56]. This problem of unobserved heterogeneity is no less true in the online purchase context, where SEMs are often used to model customers' purchase behaviors and decisions [18]. These dissimilarities are often existent but not readily observable in social data, such as consumer–behavior patterns. The situation might be particularly true while we attempt to investigate the effects of gamification affordance on online consumers' perceptions and behaviors.

In this particular context, treating all observed sets of units as belonging to a homogenous population may hide crucial differences in customer affordances and behavior [20] Referred to as the fundamental assumption of affordance, people use different objects in similar ways or the same objects in different ways based on their perception of affordances in specific contexts [16]. Because the affordance perspective draws from the relational concept of considering the interrelationships of technology with users [17], distinct classes of units (users) could possibly afford action possibilities in a different manner. We can argue that the affordance lens has the underlying assumption that distinct clusters of units may show heterogeneous behavior patterns based on their unique interaction with the technology. Different consumers may perceive different degrees of affordance offered by gamification in online shopping. For example, for one segment of shoppers, consumers' immersive experience and shopping behavior could mainly be determined by the affordance of novelty, whereas other consumers' decisions are more likely to be influenced by the affordance of rewards obtained by playing the game. Consequently, online consumers who indulge in gamification settings may exhibit dissimilar perception–behavior patterns if they belong to different classes of units.

Thus, while opting for the affordance perspective to expound a phenomenon, the question of handling the unobserved heterogeneity is not just a methodology question; rather, it is embedded in the basic tenets of the affordance perspective. Exploring unobserved heterogeneity is not only in line with the theoretical perspective of affordance but also adds valuable insights to understand a phenomenon, particularly when studied through the affordance lens. When customers do show different behavioral patterns, accounting for this unobserved heterogeneity allows targeted and more efficient strategies to be defined. However, unobserved heterogeneity cannot be captured through well-known model-analysis approaches employed normally in behavioral science such as SEM. Thereby, the method of response-based unit segmentation (REBUS) can be combined with SEM to identify the heterogeneity behavior patterns in studying gamification affordance in online-shopping behavior to bring forth in-depth valuable behavioral patterns of customers.

### 3. Research Model and Hypothesis Development

Affordance theory is embraced as the overarching theoretical foundation in the research model to interpret the impact of gamification affordances on the immersive experience and purchase intention in online shopping. This study categorizes three affordances: utilitarian affordance, social affordance, and hedonic affordance. These three affordance categories are represented as monetary rewardability, connectiveness, playfulness, and novelty. The research model is shown in Figure 1.

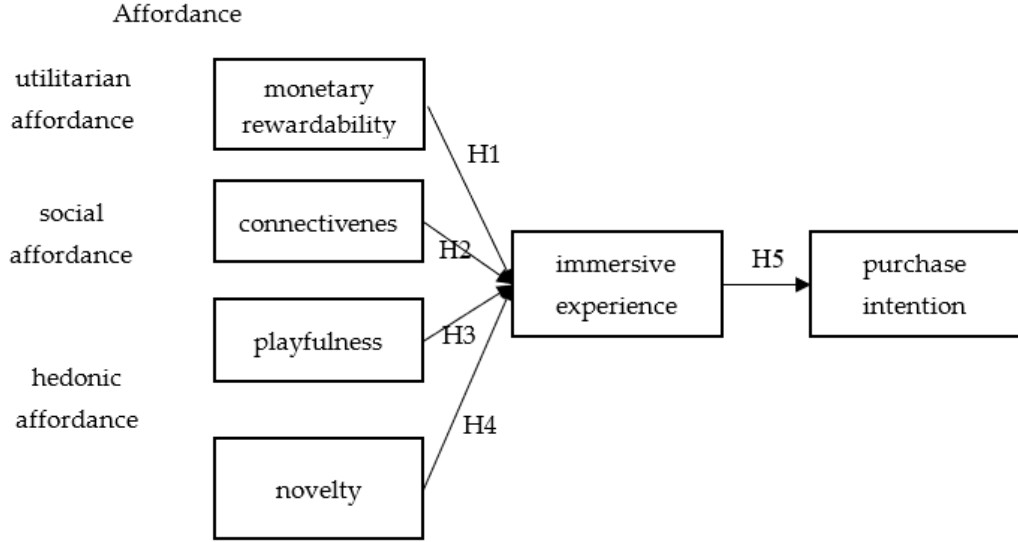

**Figure 1.** Research model.

The immersive experience is defined as a psychological state characterized by an individual's complete immersion in the activity signified by the loss of self-consciousness, intrinsic enjoyment, and a perceived sense of control over the setting emerging as a consequence of the users' interaction with gamified online shopping [57]. In other words, immersive experience describes the experience of complete involvement with satisfaction, enjoyment, and happiness in the shopping process [48,58,59]. Though the existing studies investigated immersive experiences very differently (e.g., multidimensional vs. unidimensional) based on different research purposes, researchers were generally inclined to understand immersive experience as a unidimensional construct and aimed to investigate immersive experience as a psychological state [57]. In the same vein, immersive experience is regarded as a general psychological state rather than a multidimensional phenomenon in this study

Across the literature, researchers believe that the integration of gamification can promote the experience of fun and pleasure and enhance users' experience of satisfaction and loyalty in non-game contexts [8]. When consumers are immersed in a gamification environment, they often devote full concentration and enjoy the pleasure of participating in the activities in this environment [9,35,60]. Thus, the gamified online-shopping environment can enhance consumers' immersive experience in e-commerce and make them engrossed in gamified shopping activities [61].

Moreover, shopping games have a well-designed reward mechanism to afford users diverse opportunities to immerse themselves in the process of playing games and be highly involved in the tasks to obtain the rewards [46]. First, the amount of rewards, such as coupons, free products, discounts, and virtual cash, usually increases with a higher degree of task engagement [62], such as browsing or buying the recommended products. Second, users can obtain rewards immediately by completing tasks. Prompt feedback and clear goals induce consumers to be highly involved in gaming tasks, resulting in an immersive experience. Prior studies have verified the significance of monetary rewardability affordance in a gamified context, such as its effects on flow experience in the

workplace [23] and the hedonic value of the enterprise collaboration system [46]. Thus, the affordance of monetary rewardability is a useful way to invoke users' immersion in a gamified situation, arouse their enthusiasm for completing tasks, and indulge in an immersive state. Hence, we propose the following hypothesis:

**H1:** *Monetary rewardability is positively associated with an immersive experience.*

The rise of shopping gamification generates opportunities for users to integrate social connectiveness with gaming tasks. The perceived affordance of connectiveness enables consumers to connect with others to create a team and complete the game tasks together, which is an affordance associated with satisfaction and immersion in the connective experience [15,44]. Connectiveness often elicits pleasant experiences, thus enhancing consumers' willingness to immerse themselves in this engaging environment [23]. Hence, we propose the following hypothesis:

**H2:** *Connectiveness is positively associated with the immersive experience.*

Shopping gamification provides ample gaming elements to enhance the perceived affordance of playfulness, such as playing tennis with a virtual pet and planting virtual trees with family. Lieberoth [63] suggested that gamification itself is a strategy of doing things creatively to reduce boredom and evoke continuous immersion [49,63]. The embedded game artifacts can afford the non-game system's users a game-like experience, such as playfulness, fun, and entertainment, which can facilitate an increasing immersion in the system [64]. Hence, we propose the following hypothesis:

**H3:** *Playfulness is positively associated with an immersive experience.*

Novelty does not only mean new knowledge but also specifically, new experience [65]. Individuals who experience the novelty of using an information system are likely to use it more immersively than others [66]. Embracing game elements creatively, shopping games afford consumers the possibility to have a brand-new experience, arouse consumers' curiosity [7], and inspire consumers to indulge in exploring new activities [53]. Immersing in the new experience of playing shopping games, customers are enabled to enjoy richer and more vivid touchpoints in the overall customer journal [67]. Therefore, the following hypothesis is proposed:

**H4:** *Novelty is positively associated with an immersive experience.*

Purchase intention refers to the degree to which consumers intend to buy a product in the context of gamification in e-commerce. This study intends to investigate the mechanism of gamification in a general situation of an e-commerce platform. The observation of the phenomenon suggests that the gamification strategies applied on the e-commerce platform remain the same across different products and different stores. In other words, the basic gamification mechanism remains the same across the stores and products on this platform. Since the primary purpose of this research is not to explore how the gamification mechanism impacts the purchase intention towards different stores or products differently, we investigated the consumer's general purchase intention.

The mechanism of gamification in online shopping exposes consumers to marketing promotions during the gaming process. Immersive experience can facilitate users' perception of both the store and available products by providing them with an innovative customer journey [68]. In addition, while indulging in completing gaming tasks, consumers may develop positive evaluations of products recommended easily. For example, gaming tasks include browsing agriculture products from poverty areas. Customers are likely to contribute to poverty alleviation by buying these products. Prior studies have shown that immersive experiences can induce a higher degree of satisfaction and involvement, which plays a cardinal role in influencing sales and store loyalty [69]. Hence, the following hypothesis is proposed:

**H5:** *Immersive experience is positively associated with purchase intention.*

## 4. Methodology

The primary purpose of this study was to examine the effect of gamification affordances in the global research model proposed in our research context and then explore the potential heterogeneity in the global research model based on the affordance perspective to offer finer-grained results. In other words, we created an analogy of an unobserved heterogeneity analysis to a post hoc analysis of a study. The unobserved heterogeneity analysis was a natural extension of the primary research purpose and aimed to add more knowledge and understanding with a two-step research design.

First, we examined the global research model with empirical data and verified the role of gamification affordance in influencing the dependent variables. Second, addressing the fundamental assumption of affordance that people may have different perceptions of gamification affordances, we employed REBUS to identify the heterogeneity behavior patterns in our research context to unveil richer and more detailed results beyond the common confirmatory factor analysis (CFA) towards the global model. This REBUS analysis moved one step further to unveil how affordances are perceived differently and thus bring forth in-depth valuable behavioral patterns of different consumers. We endeavored to make these two steps closely connected and coherent both in the literature review section (Section 2) and data-analysis section (Section 5). We believe the results from these two steps may jointly offer a more comprehensive explanation of the phenomenon.

### 4.1. Measurement Development

We developed the corresponding measurement scales of our model by adopting measurements from previous literature. A 5-point Likert scale was adopted in the survey, ranging from (1) strongly disagree to (5) strongly agree. The scales for measuring monetary rewardability were adopted from the work of Suh et al. [23]. Adapting the scales of prior work [47], we revised the scales of connectiveness. The measurements of playfulness and novelty were developed by Feng et al. [45] and McLean [70], respectively. In addition, the scales from the work of Daassi and Debbabi [71] and Zhang et al. [12] were modified for an immersive experience and purchase intention, respectively. The items are shown in Table S2 in the Supplementary Material. All of the constructs used in this research are reflective constructs.

This study conducted expert interviews to improve the quality of the survey [11]. Four interviewees were practitioners in gamification in e-commerce and six interviewees were professors in this area. They were interviewed to provide professional suggestions to further validate the questionnaire. We finalized the survey to address the issues indicated by these 10 experts, such as minor modifications of wording and sentences to improve readability. Eventually, all the experts believed the questions were proper and easy to understand.

### 4.2. Data Collection

This study used an online survey platform (Sojump) to gather data for empirical analysis. Sojump is the most influential online survey site. One hundred and seventy-eight million questionnaires have been published and 13.913 billion questionnaires have been collected on this website [72]. This platform provides access to a large population of the target sample, thus reducing social-desirability bias through random sampling [73].

The questionnaires were distributed from 5 July to 15 July 2022. To encourage participation, we provided the respondents with a small amount of money. Only people with experience in gamification online shopping on Taobao were invited to answer the questions, as Alibaba is one of the most influential e-commerce giants in the world, with billions of users all over the world. Gamification has been integrated widely into Alibaba's ecosystems (such as Taobao), as well as its daily operation and marketing activities. We believe Alibaba can present the mainstream application of gamification in e-commerce. In addition, though different games have been designed, such as playing with chickens, feeding a kitten, planting a tree, Candy Crush, and a treasure hunt in the ocean, the basic mechanism remains

the same, such as the possibility of seeking novelty (e.g., experiencing both playing and buying), obtaining playfulness and rewards (e.g., browsing or buy products to earn fodder or fertilizer), and socializing during the shopping process (organizing a game group with friends) [15]. Hence, Taobao offers a good representative situation for this study to explore the general mechanism of gamification affordance in e-commerce.

Attention-check questions such as opposing questions were set in the questionnaire to identify and screen out the respondents who were not paying enough attention. We set one opposing question of playfulness that changed the direction of the scale in the questionnaire (PLY4: Gamification in online shopping is not fun for me). A data-screening procedure helped us to identify valid data based on three standards: (1) All the questions must be answered, (2) the respondent must pass the attention-check questions, and (3) respondents should not give the same score to all questions.

Finally, of the 243 questionnaires, 212 were valid (87.2%) for the final data analysis. In order to ensure we had a sufficient sample size, an independent-sample t-test was conducted with the help of G*Power v3.1.9.4. [74,75]. With an anticipated effect size of 0.2, a sample size of 68 was required to achieve a statistical power of 0.95. The result of the G*power test implied that the hypothesis analysis was based on sufficient statistical power to detect relational effects. The demographic information of the respondents can be found in Table S3. Among the valid samples, 65.6% were female and 34.4% were male, which is roughly consistent with Lin [76]. Similar to Al-Zyoud [37], most of the respondents in our study were under the age of 30 (91.5%). As for educational level, the respondents had mainly obtained bachelor's degrees or above, which accounted for 87.2% of the total and is similar to Hsu and Chen [33]. In terms of gamification experience, 27.4% had experienced it for less than a year, about half of the respondents for 1–2 years, and the others for more than 3 years.

In conventional PLS-SEM analysis, homogeneity of the data set is assumed. However, the recent guidelines on PLS-SEM question this homogeneity assumption and contend that in most cases the dataset is composed of heterogeneous groups. Both observed and unobserved factors can be sources of heterogeneity in a data set. When the heterogeneity in the sample is caused by observable factors such as age, income, gender, race, etc., generally a multi-group or moderation analysis is performed on the data to analyze the effects of observed heterogeneity [18]. In the modern PLS guidelines, it is argued that heterogeneity is not only caused by observable factors but also by some unobservable causes [77]. Thus, unobserved heterogeneity cannot be ignored and the universality of the assumption of homogeneity of data cannot be maintained in PLS analysis. Unobserved heterogeneity can be analyzed through various approaches. The two most popular ones are the finite-mixture partial-least-squares (FIMIX-PLS) method [20] and REBUS-PLS.

Unit segmentation is PLS path modeling that can be detected with the employment of REBUS-PLS [78]. The present study employed the REBUS-PLS method instead of FIMIX for three reasons. One, REBUS-PLS is a response-based method employing an iterative algorithm that is highly suitable for the assessment of heterogeneity in predictive SEM [79]. Two, the FIMIX method can detect heterogeneity only in the structural model, whereas REBUS-PLS is capable of considering heterogeneity in both the measurement and structural models. Three, REBUS-PLS is a distribution-free technique, meaning it does not require the fulfillment of the condition of normality of data [78]. One of the key features of REBUS-PLS is that it cannot only estimate the unit membership of each respondent in latent classes but also estimate class-specific parameters of each local model based on latent class members simultaneously [80]. Because of its utility, REBUS-PLS is famous among scholars for analyzing heterogeneity in data [81].

### 4.3. Common Method Variance (CMV)—Construct-Level Correction (CLC) Approach

To minimize the influence of common method variance (CMV), this study employed construct-level correction (CLC) approach under the measured latent marker variable (MLMV) technique, which is the most appropriate approach for controlling for CMV under

PLS-SEM [82]. In the CLC approach, CMV control constructs are created equal to the number of constructs in the research model, and each CMV control construct is modeled to impact one construct in the actual PLS model [83,84]. Each CMV control construct uses the same entire set of measured latent marker variables, and then path coefficients are estimated again after introducing CMV control constructs [85]. The obtained path coefficients of constructs with and without the CMV control constructs are presented in Table S4. $\beta_0$ are path coefficients of constructs with the CMV control constructs and $\beta1$ are path coefficients of constructs without the CMV control constructs. Table S4 also presents the significance values of path coefficients with and without CMV control constructs. It can be observed that introducing CMV control constructs did not alter the significance of even a single coefficient. As a final check, the differences in the coefficients' values are also presented in Table S4, and these changes were very small and not significant. Apart from the coefficients' values, the construct-level correction (CLC) approach [82] also recommends studying the influence of CMV control constructs on $R^2$ values. It is important to note that any significant difference between the CLC estimation and PLS estimation would be the indication of CMV in the study, whereas no significant difference between CLC estimation and PLS estimation would mean no potential impact of CMV on the study's results [85]. The effects of introducing the CMV control constructs on path coefficients and $R^2$ were very small; besides, these changes were not significant, which indicates that common method variance was not a serious issue in this study.

## 5. Analysis Results

### 5.1. PLS-SEM (Global Model) Results

The global model (Figure 1) was examined across the full sample using the PLSPM package version [86] in R-Studio [87], following the confirmatory factor analysis (CFA) approach. The goodness-of-fit (GoF) value of the global model was 0.562. In modern PLS-SEM, internal consistency reliability is measured through Cronbach's alpha and Dillon–Goldstein's ρ [88]. As indicated in Table S5, for all the constructs, the values for both these indicators were larger than the threshold of 0.7 [89,90]. Table S5 illustrates that the standardized loadings for all the items in the global model were above the threshold of 0.7 [90] and bootstrapping t-values were significant for all the loadings, thus ensuring indicator reliability. Another measure for consistency is the unidimensionality of constructs, which is measured through eigenvalues [91]. All the constructs showed unidimensionality, as the first eigenvalue was greater than one and the second eigenvalue was less than one for all the constructs [88,91]. Convergent validity for the global model was established, as all the constructs had AVE values > 0.5 [90]. Discriminant validity for the global model, following Hair et al. [88]., was demonstrated through the heterotrait–monotrait ratio of correlations (HTMT) and the Fornell–Larcker criterion in Tables S6 and S7, respectively. All the values for HTMT were below the threshold values of 0.9, and for the Fornell–Larcker criterion, all the squared inter-construct correlations were below the AVE values for each construct. Both these criteria indicate sufficient discriminant validity in the global model [92,93].

The following section presents the results of the structural (inner) global model in Table 1. Out of five proposed hypotheses, four (H2, H3, H4, and H5) were statistically significant at the 95% confidence level. As per the standardized loadings and path coefficients shown in Table 1, only the hypothesized path coefficient between connectiveness and the immersive experience was slightly weak with significant at the 95% confidence level. The overall path-model results and bootstrapping-significance values reflected a correctly conceptualized path model. Among all the relationships, the strongest relationship was between immersive experience and purchase intention, with a path coefficient of 0.56. In terms of the interrelationship between affordances and immersive experience, monetary rewardability affordance ($\beta = 0.44$) appears to have had the strongest influence on the immersive experience in the global model, whereas hedonic affordance came in second with $\beta = 0.24$.

**Table 1.** Estimated path coefficients.

| Path | Estimate | Std. Error | t-Stat. | *p*-Value | 95% Confidence Interval | |
| --- | --- | --- | --- | --- | --- | --- |
| | | | | | Lower | Upper |
| Connectiveness~Immersive experience | 0.093 | 0.050 | 1.980 | 0.049 | 0.0015 | 0.1976 |
| Playfulness~Immersive experience | 0.239 | 0.066 | 3.602 | 0.000 | 0.1166 | 0.3699 |
| Novelty~Immersive experience | 0.189 | 0.061 | 3.079 | 0.002 | 0.0633 | 0.3043 |
| Monetary rewardability~Immersive experience | 0.437 | 0.053 | 8.257 | 0.000 | 0.3332 | 0.5398 |
| Immersive experience~Purchase intention | 0.562 | 0.047 | 12.072 | 0.000 | 0.4653 | 0.6486 |

The following Table 2 presents R-squared values with bootstrapped significance results. The global model explains the 32% variance in purchase intention and the 56% variance in the immersive experience. The path relationship between immersive experience and purchase intention depicts a large effect size ($f^2$) with a value of 0.46.

**Table 2.** R-square and effect size.

| | $R^2$ | Std. Error | 95% Confidence Interval | | | $f^2$ | |
| --- | --- | --- | --- | --- | --- | --- | --- |
| | | | Lower | Upper | | | |
| Immersive experience | 0.563 | 0.0476 | 0.481 | 0.665 | Immersive experience→ | 0.462 | Large |
| Purchase intention | 0.316 | 0.0509 | 0.222 | 0.421 | Purchase intention | | |

*5.2. REBUS-PLS (Local Model) Results*

In the next stage, to evaluate the effects of unobserved heterogeneity, the REBUS-PLS approach was applied. The presence of unobserved heterogeneity gave rise to latent homogenous segments in the dataset, represented through the "local" or "class-specific" models [94]. REBUS was performed on the proposed global model with a full dataset. The results indicated the presence of unobserved heterogeneity with three distinct latent classes. The group-quality index (GQI) value for the three-class solution was greater than the GOF value of the global model, indicating a three-cluster solution. In addition, the G*power test results illustrated that no more than three classes could be obtained with a sufficient sample size, and the GOF values for the three-class solution was greater than both two-class and one-class solutions. Finally, the GQI value for the class 3 solution was greater than the class 1 and 2 solutions. Thus the three-class solution was deemed most appropriate in light of the REBUS guidelines. The cluster-analysis dendrogram for the global model (see Figure 2) also suggested that a three-class partition was acceptable. Moreover, AVE for all the constructs in the three-class solution was above the required 0.5 thresholds, and Dillon–Goldstein's ρ exceeded the threshold of 0.7, providing further support for the reliability and validity of the REBUS-PLS three-class solution [89,95,96]. The results of the path analysis of the three classes are presented in Table 3.

**Table 3.** REBUS-PLS results.

| | Coefficients | | | | *p*-Values | | | |
| --- | --- | --- | --- | --- | --- | --- | --- | --- |
| Path | Global Model | LM1 | LM2 | LM3 | Global Model | LM1 | LM2 | LM3 |
| Connectiveness~Immersion | 0.093 | 0.3063 | 0.2678 | −0.0154 | 0.060 | 0.002 | 0.0001 | 0.8424 |
| Playfulness~Immersion | 0.239 | 0.2675 | −0.1239 | 0.3644 | 0.000 | 0.020 | 0.1374 | 0.0001 |
| Novelty~Immersion | 0.189 | −0.039 | 0.2432 | 0.2106 | 0.002 | 0.716 | 0.0005 | 0.0241 |
| Rewardability~Immersion | 0.437 | 0.4284 | 0.6246 | 0.4397 | 0.000 | 0.000 | 0.000 | 0.000 |
| Immersion~Purchase intention | 0.562 | 0.702 | 0.9259 | 0.9129 | 0.000 | 0.000 | 0.000 | 0.000 |

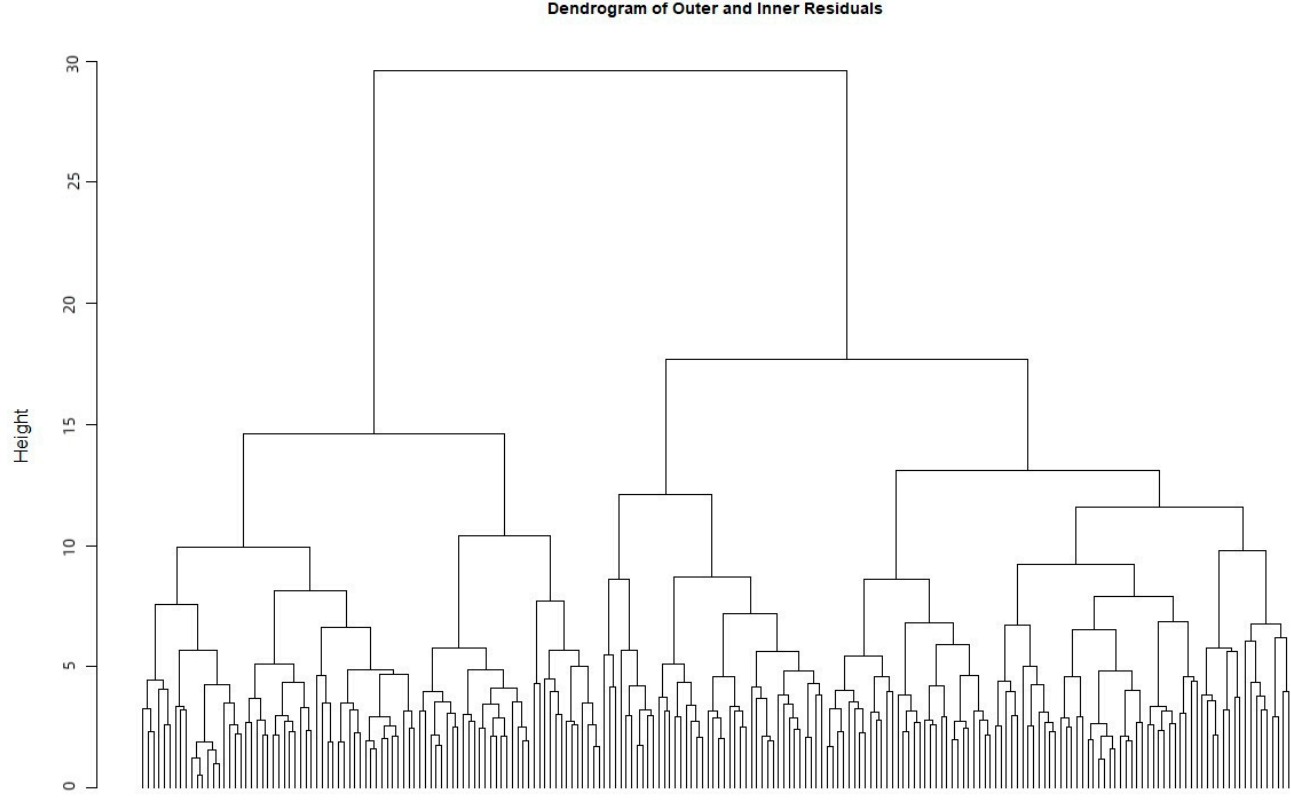

**Figure 2.** Dendogram obtained by cluster analysis on the residuals from the global model.

Class 1 (*n* = 71): The path coefficients from monetary rewardability (utilitarian affordance), playfulness (hedonic affordance), and connectiveness (social affordance) to immersion reflected significant effects. The strongest effect on immersion was from monetary rewardability ($\beta$ = 0.428). In this class, contrary to the global model, connectiveness had a very strong and significant effect on immersion, with a path coefficient of 0.306 and *p*-value < 0.05. However, in this class, novelty affordance had no impact on immersion. The link between immersion and purchase intention was strong, with $\beta$ = 0.702. Since in this class only the effect of novelty affordance was insignificant, the respondents in this class were labeled "no novelty" users. For class 1 the path link between connectiveness and immersion showed a regression coefficient of 0.306, which was greater than those obtained for the second and third classes. For class 1, the GOF and $R^2$ (purchase intension) values were 0.575 and 0.493, respectively, which were greater than those for the global model.

Class 2 (*n* = 69): The second class was characterized by significant path coefficients between all affordances and immersion except playfulness. Playfulness affordance, in this class, had an insignificant relationship with immersion. However, similar to the other two classes, in this class the path between monetary rewardability affordance and immersion was the strongest among all affordances. Thus, immersion in this class was driven by the absence of playfulness elements. We labeled the respondents in this class as "no playfulness" users. It is also interesting to note that the path coefficient between immersion and purchase intention for this class ($\beta$ = 0.926) was strongest among all the classes, as well as stronger than the global model. The GOF value for this class (0.738) was greater than for the global model, which was a prerequisite for a class to be considered satisfactory.

Class 3 (*n* = 72): This is the only class that demonstrated no significant relationship between connectiveness and immersion, which is similar to the results of the global model. Immersion in this class was driven by monetary rewardability, playfulness, and novelty affordances. This class depicted the most significant path coefficient, linking immersion and playfulness with β = 0.364. Like the other two classes, in this class monetary rewardability affordance impacted the immersion more than other affordances. Besides, the regression coefficients for monetary rewardability to immersion were strongest in this class, with β = 0.625, compared to the other two classes. As a result, we can say that in this class, users were more sensitive to monetary rewardability factors compared to other classes. The relationship between purchase intention and immersion in this class was also very strong (β = 0.913) and similar to class 2, as connectiveness affordance is the only affordance that did not significantly affect immersion in this class. This class was labeled as "no connective" users. The GOF value for class 3 was 0.755, which was greater than that of the global model.

### 5.3. MGA

Based on Rebus-PLS's three-class solution and respective class segmentation, an MGA analysis was run. Before running the traditional MGS in PLSPM to determine the path-by-path estimate differences, following the approach of Klesel et al. [77] an overall model-difference assessment was performed based on the squared Euclidean-distance (dE) and the geodesic-distance (dg) tests by comparing the model-implied indicator-correlation matrix across groups [77]. The MGA results based on this latest method in PLS-PM also supported the three-class segmentation of the dataset. Both tests depict a significant difference in the overall model among all three latent classes. After the overall model assessment, we performed a path-by-path estimate analysis to examine the individual path difference among all three heterogeneous groups. For the sake of robustness, two separate methods were employed to test the path-by-path difference between the groups. MGA test was conducted to calculate the group-wise parameter differences simultaneously. The second was a non-parametric test based on a confidence-interval approach drawing significance between the group differences based purely on bootstrapping [97,98]. The detailed results of path-by-path MGA with the difference between path coefficients and their respective significance values (*p*-values) are present in Table 4.

(1)   The effects of connectiveness affordance on immersion did not differ between class 1 and class 2, whereas the difference was significant between class 1 and class 3. Class 2 and class 3 also differed in terms of connectiveness affordance → immersion.

(2)   Because class 2 was the only class not affected by playfulness affordance, the effect of playfulness affordance on immersion between class 1 and class 2 as well as between class 2 and class 3 was statistically different. The path coefficients of class 1 and class 3 for playfulness affordance to immersion relationships were not different from each other, as both classes provided significant importance to playfulness factors (Table 4).

(3)   For novelty affordance, MGA analysis revealed a statistical-path difference only between class 1 and class 2. For other class comparisons, there were no differences.

(4)   As observed in REBUS-PLS results, rewardability affordance turned out to be the strongest factor while determining immersion. For all three latent classes, rewardability affordance was a very strong influencer of immersion. This finding was further substantiated by the MGA results, which showed that the path coefficients between rewardability affordance and immersion were almost equally strong among all the classes, reflected as "no difference between the path coefficients among all the classes." To conclude, the results from the MGA path-by-path analysis further validated the three-class solution obtained from REBUS-PLS, thereby supporting the notion of unobserved heterogeneity in PLS-SEM.

**Table 4.** Multi-group analysis results.

| Groups | Difference | t-Value | *p*-Value | Significant |
|---|---|---|---|---|
| Immersion~Connectiveness | | | | |
| 1 vs. 2 | 0.0385 | 0.3225 | 0.7476 | No |
| 1 vs. 3 | 0.3217 | 2.5968 | 0.0105 | Yes |
| 2 vs. 3 | 0.2832 | 2.7316 | 0.0071 | Yes |
| Immersion~Playfulness | | | | |
| 1 vs. 2 | 0.3914 | 2.782 | 0.0062 | Yes |
| 1 vs. 3 | 0.0969 | −0.6536 | 0.5145 | No |
| 2 vs. 3 | 0.4883 | −3.872 | 0.0002 | Yes |
| Immersion~Novelty | | | | |
| 1 vs. 2 | 0.2822 | −2.2293 | 0.0277 | Yes |
| 1 vs. 3 | 0.2496 | −1.77 | 0.079 | No |
| 2 vs. 3 | 0.0326 | 0.2825 | 0.778 | No |
| Immersion~Rewardability | | | | |
| 1 vs. 2 | 0.1962 | −1.6834 | 0.0947 | No |
| 1 vs. 3 | 0.0113 | −0.0832 | 0.9338 | No |
| 2 vs. 3 | 0.1849 | 1.5059 | 0.1346 | No |
| Purchase intention~Immersion | | | | |
| 1 vs. 2 | 0.2239 | −3.2006 | 0.002 | Yes |
| 1 vs. 3 | 0.2109 | −3.0063 | 0.0035 | Yes |
| 2 vs. 3 | 0.013 | 0.4525 | 0.6516 | No |

## 6. Discussion

The affordance perspective was employed to study user behavior in an established practice in the relevant literature [13,15]. This study extended the conceptual model to interpret how affordance-backed immersion leads to final purchase intention in the e-commerce gamification environment. PLS-SEM analysis was used to empirically assess the hypothesis in the global model involving the full dataset. The analysis found that the proposed affordances in gamification (playfulness, monetary rewardability, novelty) were significant and positively related to user immersion. Only one affordance in the global model had a slightly weak effect and was significant at the alpha level of 6% instead of the usual 5%. Overall, the effect of affordances on user behavior as established in our model has been well recognized in the literature [44]. In line with our findings, Suh et al. [23] suggested that monetary-rewardability affordance was a strong predictor of immersion. Hamari et al. [64] observed that playfulness elements played a crucial role in determining immersion. The findings of this study are in line with the results of Shi et al. [7] and suggest that a significant relationship exists between novelty and immersion. Through the full sample analysis, a significant linear relationship between immersion and purchase intention in the gamification environment was also supported, which has also been validated by Erdmann et al. [99] and Pantano and Servidio [100].

However, while analyzing our dataset for heterogeneity by employing the REBUS-PLS approach, we identified that our dataset had three distinct classes/segments instead of a homogenous sample. Focusing on the variability in relationship strength between gamification affordances and immersion, REBUS-PLS analysis resulted in richer and deeper insights into e-commerce gamification users. The results revealed that instead of a single homogenous set, gamification users fall into three distinct classes based on their response patterns. One of the most crucial insights was that path coefficients, as they appear in the global model, are markedly different when compared with individual latent-class path coefficients. For instance, the global model depicted a very weak path coefficient (β = 0.09) between connectiveness affordance and immersion; however, for class 1, the same path coefficient was 0.30. Similarly, the beta coefficient between playfulness affordance and immersion in the global model was a very strong 0.24, but in class 2 the same beta was insignificant (as good as zero). These are huge differences and reveal an important insight, i.e., if there is heterogeneity in the sample set, the results of the global model can be very misleading.

More strikingly, the same level of discrepancy was observed in terms of the model's explanatory power. By looking at the global model's results we can conclude that the model explained 56% of the variance in immersion and 32% of the variance in purchase intention. However, when we observed the variance explained in the three latent-model solutions based on the segmentation of REBUS-PLS, we saw that for class 2 the variance explained was over 80% both for immersion and purchase intention. Similarly, in class 3 the $R^2$ was over 80%. Thus, the three-class segmentation solution fit the data multiple times better than the global model, which assumed no heterogeneity. By providing a detailed comparison between the global and local models' results with the help of REBUS-PLS and by clearly explicating how global model's results can be misleading in the presence of heterogeneity, our research highlights the importance of examining unobserved heterogeneity in PLS-SEM modeling and collaborates with the arguments presented in recent PLS-SEM literature [19,20].

In addition, REBUS-PLS also provides useful information regarding the crucial differences between the members of each class. For instance, it was quite apparent that not all affordances affected all classes equally. Not all respondents drew affordances from the gamification phenomenon in a similar manner. For class 1 ($n = 71$), a significant positive relationship between playfulness, monetary rewardability, and connectiveness affordances and immersion was observed (thus, H1, H2, and H4 were supported for this class of respondents). For class 2 ($n = 69$), the results depicted a positive association between monetary rewardability, novelty, and connectiveness affordances and immersion with statistical significance (thus, H1, H3, and H4 were supported for respondents belonging to class 2). As for this particular class of respondents, we could not find support for H2 since the association between playfulness affordance and immersion was not significant. For class 3 ($n = 72$), the analysis showed a significant positive relationship between playfulness, novelty, and monetary rewardability affordances and immersion (thus, for latent class 3 support was found for H2, H3, and H4), but the relationship between connectiveness affordance and immersion remained unsupported.

## 7. Contributions and Limitations

### 7.1. Theoretical Implications

Generally speaking, this study aimed to provide theoretical implications from four perspectives. First, applying affordance theory to study a novel phenomenon, this study contributes to both the literature on e-commerce and affordance theory in two ways. On the one hand, as one of the first to apply affordance theory in studying gamification in an online-shopping environment, this study extends the generalizability of the affordance perspective in a novel context. On the other hand, this study provides empirical evidence to identify three classifications and the specific representative affordances for gamification on online-shopping websites. The prior approach of identifying specific affordances in a particular context without general classification may offer little opportunity for comparison of results, such as specific affordances in the social-commerce context [48,76] or different gamification affordances in crowdsourcing [45] and enterprise collaboration systems [46]. Hence, this study not only contextualizes the affordance perspective to interpret consumer behaviors in a gamification-embedded shopping environment but also offers new insights regarding the representative affordances in each classification and thus enables the possibility to accumulate knowledge and evidence for future comparison.

The second implication addresses immersion as the critical outcome of gamification settings and reveals the relationships between affordances and immersion in gamification-embedded online-shopping environments. Although several studies have explored the actions facilitated by contextual affordances [12,70], little research has explored the linkage between affordance and immersion. By exploring this relationship, this study interprets the consumer decision-making process enabled by gamification affordances via immersion.

The third contribution arises from a conceptual perspective regarding the underlying notion of affordance. Utilizing REBUS-PLS to analyze a global model and heterogeneous sub-groups, this study provides an in-depth understanding of empirical evidence regarding

the heterogeneous nature of affordance. The results confirm that people who belong to different classes of units may show heterogeneous behavior patterns regarding the perception of a specific affordance. The examination of unobserved heterogeneity in this study detected three distinct latent classes of e-commerce gamification users with different response patterns in terms of drawing affordances for an immersive experience, confirming the presence of unobserved heterogeneity.

The fourth contribution of this study is from a methodological perspective. From a procedural perspective, it adds to the recent SEM-PLS literature and REBUS-PLS literature. Our study specifically examines the effects of heterogeneity in the research sample, which is declared an essential element to check the correctness and robustness of PLS path modeling [20,101]. Though SEM scholars have pointed out on various occasions that ignoring the unobserved heterogeneity can lead to misleading conclusions and wrong statistical estimations [19,20,102], in business research it is still often ignored [56,103]. Thus, our findings highlight the importance of unobserved heterogeneity and substantiate the necessity of conducting an unobserved heterogeneity analysis in a PLS-SEM study [19,102], which emphasizes the fact that more attention should be given to latent classes that may exist in a dataset.

### 7.2. Practical Implications

In addition to the practical implications, this in-depth analysis of affordance in gamification-embedded online-shopping environments offers unique insights for practitioners and developers as well. The findings highlight the gamification affordances that lead to a high-quality immersive experience that subsequently solidifies purchase intentions. Thus, the developers of e-commerce platforms should target those specific features of gamification, such as gamification factors that offer users the opportunity to obtain rewards (rewardability affordance).

Furthermore, apart from the specific affordances, this study unveiled that latent classes of users have different response patterns towards gamification affordances in the e-commerce gamification environment. The distinctive characteristics of each latent class reflect that a more specific and appropriate immersive environment should be developed catering to the needs of each specific group. For instance, latent class 3 had no use for social affordance, and the strongest relationship in this class to the immersive experience was from playfulness. To keep this segment of users engaged, the e-commerce platform employing gamification must ensure a high level of playfulness.

Last, taking unobserved heterogeneity into consideration offers more specific determinants of immersive experiences based on the distinct features of each latent class. It is recommended that platform developers understand that gamification elements should be diversified enough to fulfill the specific requirements of each latent class. Only then can e-commerce platforms foster the desired level of immersive user experience and purchase intention. It is pertinent to note that these deeper insights were possible due to the REBUS-PLS technique. To conclude, the findings of this study offer unique and valuable insights for practitioners and developers, on the one hand, related to the mechanism of how gamification in online shopping affords users opportunities for purchase decisions, and on the other, about the potential variation in behavior of different segments of users involved in gamified e-commerce.

### 7.3. Limitations and Future Studies

Apart from the meaningful contribution, this study is bound by its limitations, which provide opportunities for future research. Firstly, this study considered a broader view of the affordance perspective, and the affordances mentioned in the presented study cover the overall gamification scenario in e-commerce. Although this broader affordance perspective provides deep insights regarding e-commerce user behaviors, we cannot offer an exclusive list of all the possible constructs and classify them according to the three types of affordances. Future studies can explore more contextualized affordances from

specific affordance perspectives, such as the technology-affordance perspective, which can offer additional insights and bring forth technology affordances that are specific to the gamification context.

Secondly, the research only investigated the mechanism of gamification in a general situation of an e-commerce platform and did not study a specific type of gamification. Literature on gamification shows that different gamification has distinct features, which may afford users distinct opportunities for actions, and thus may lead to different types of affordances. Future researchers are encouraged to study subtypes and more specific types of e-commerce gamification. Third, even though we ran a thorough multi-group analysis, the moderation–interaction effects of variables were not tested in this research. We believe that testing these interaction effects can lead to more useful insights. In addition, our empirical data were collected from the consumers experiencing the gamification integrated on Taobao, which may have restricted the generalizability of the results. Future studies should endeavor to explore more applications of gamification in e-commerce. Finally, this study, like all survey-based studies, may have been affected by self-report bias. The relatively low values of standard deviation (SD) may indicate that most of the respondents selected neutral statements or slightly positive ones. Though we have tried to control for it by following the best practices of survey design [57,104], we encourage future studies to employ other research designs like qualitative–quantitative mixed methods or experimental research to validate the findings of this study.

**Supplementary Materials:** The following supporting information can be downloaded at: https://www.mdpi.com/article/10.3390/jtaer18010016/s1, Table S1: Literature review of specific affordances identified in prior literature; Table S2: Measurements; Table S3: Demographic distribution (N = 212); Table S4: CMV-CLC results; Table S5: Reliability and Convergent Validity; Table S6: Heterotrait-monotrait ratio of correlations matrix (HTMT matrix); Table S7: Fornell-Larcker matrix.

**Author Contributions:** Conceptualization, S.M.U.T. and X.-Y.X.; methodology, S.M.U.T. and X.-Y.X.; software, S.M.U.T.; validation, S.M.U.T. and X.-Y.X.; formal analysis, S.M.U.T.; investigation, X.-Y.X. and Q.-D.J.; resources, Q.-D.J.; data curation, S.M.U.T.; writing—original draft preparation, X.-Y.X., S.M.U.T. and K.W.; writing—review and editing, X.-Y.X., S.M.U.T., Q.-D.J. and K.W.; visualization, S.M.U.T.; supervision, X.-Y.X.; project administration, X.-Y.X. All authors have read and agreed to the published version of the manuscript.

**Funding:** This research was funded by Shaanxi Natural Science fundamental Research Program [grant number: 2023-JC-QN-0794].

**Institutional Review Board Statement:** Not applicable.

**Informed Consent Statement:** Informed consent was obtained from all subjects involved in the study.

**Data Availability Statement:** Not applicable.

**Conflicts of Interest:** The authors declare no conflict of interest.

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
