# Peer review of "Exploring the Gamification Affordances in Online Shopping with the Heterogeneity Examination through REBUS-PLS"

_jtaer, doi:10.3390/jtaer18010016_

Round 1

Reviewer 1 Report

Review Comments

Title: Examining heterogeneity through REBUS-PLS: An affordance perspective of gamification in online shopping with predictive modeling

The manuscript presents good research in describing a relatively unexplored phenomenon of gamification through a unique perspective of affordance and an exploratory approach to REBUSPLS. The topic of the manuscript is interesting from the theoretical perspective and important in practical terms. Gamification applications, such as its application in e-commerce, have become very popular. However, I think how the consumers are empowered by such novel technology or unique features has received relatively little attention in prior research. It is encouraged that the authors of current paper applied REBUS-PLS to not only explore the heterogeneity in general SEM analysis, but also build the theoretical connections between heterogeneity and affordance theory. I believe it creates certain significant contribution to this field. Prior literature, including the original papers that discussed the affordance concept, has clearly stated the heterogeneity of affordance. However, few studies have examined it using empirical tools. I believe this manuscript has its novelty. Hence, this study can provide useful knowledge since it uses the affordance perspective as a good example of technology empowered decision-making processes and behaviors. 

Overall, the manuscript is very engaging and easy to follow. As a result, I am to recommend that the manuscript can be published in the journal with minor revisions (some minor suggestions are listed below). There are a few minor issues, which I would recommend the authors to address them before the publication. Since they are Minor ones, they won’t prevent the publication of the paper.

1. General comments (Section 1 & 2). There are no major issues in the introduction (Section 1) or the literature review (Sections 2) of the paper. The introduction describes in adequate detail the motivation and objective of the study, and the literature review seems to go through the relevant prior research on affordance in E-commerce or gamification. But, I would suggest the authors delete Table 1 from the manuscript, but please move it into the appendix. The current version is relatively long, and I think my suggestion can improve the readability.

2. In terms of model development (Sections 3), the authors do a good job of arguing why the affordances identified are positively associated with an immersive experience in gamification in online shopping.

3. General comment (section 4.1-4.2). In terms of methodology (Section 4), all the measurements seem to be based on well-established scales from prior studies and were also tested appropriately before their use and there are also no obvious issues in data collection. Although I have a few minor concerns about other parts.

4. Comment 4-3. It is mentioned in the manuscript in paragraph 2 in section 4.2 that “the respondent must pass the attention check questions such as repeating and opposing questions ". I think the readers will understand more details if the authors can describe in more details about the attention check questions as well as how they were used.

5. Comment 5-2. Please mention that you were using reflective rather than formative measures.

6. Comment 5-3. Please mention in your results section that you are reporting the standardized loadings and path coefficients.

7. General comment (section 6 & 7). In the discussion (Section 6), the results of the study are summarized and reviewed in adequate detail; several interesting and important theoretical and practical implications are presented. Overall, these implications seem valid (section 7), although I have several minor notes. Comment 6-1. I understand that the authors intended to present more practical implications based on their results, but the current version is relatively long. Thus, I suggest the authors to focus on discussing the most significant and critical practical implications.

Reviewer 2 Report

Dear authors, I hope my comments will be useful and will let you improve the article.

The review of the article “Examining heterogeneity through REBUS-PLS: An affordance perspective of gamification in online shopping with predictive modelling”

This study investigates the impact of gamification on purchase intention through a mediating variable – immersive experience and use of REBUS-PLS to show the nature of heterogeneity in perceived affordances and ensure the robustness of structural model results. I found the article quite interesting and I hope that my comments will let the authors make findings more important to the science. Here are several major shortcomings.

1. The article is twofold. One part is related to the analysis of consumer behavior – it tries to prove the impact of rewardability, connectiveness, playfulness, and novelty seeking on purchase intention. Another part proves the usefulness of REBUS-PLS for the analysis of unobserved heterogeneity. All reviews of literature and hypotheses are dedicated (mainly) to the first part of the article, while the title tells just about the second part of it. Therefore there is a lack of congruence among the different parts of the article.

2. Theoretical part presents a lot of information about gamification and affordance, but it has almost nothing on an immersive experience, which must be a key element in the model because all independent variables have no direct impact on purchase intention. The same happens almost with all variables in the model since the theoretical part tells a lot about utilitarian, social and hedonic affordance. That rises a question – why are some affordances represented by one variable, while hedonic – by two? Can utilitarian affordance be represented just by rewardability or ease of use could be a part of it as well? Can some rewards be hedonic and rewardability would differ depending on statements that were used for measurement of it? The same happens with social affordance. Maybe it is just a part of hedonic affordance, since relations bring hedonic values. Therefore the theoretical part must be rewritten.

3. Methodological part does not tell anything about the used gamification – its elements, system of elements, dynamics, etc. Probably it would illogical to expect a high level of immersive experience if participants simply had to remember their experience in online shopping.

4. Another major issue is related to measurements. Purchase intention was measured using statements about impulse buying behaviour. The statements did not measure either the purchase intention of a certain product or the purchase intention in a certain store they are about purchasing any unknown product or quantity of unknown products, or even trait. The same problem exists with the measurement of immersive experience, which is a multidimensional phenomenon (Zhang, C. (2020). The why, what, and how of immersive experience. IEEE Access8, 90878-90888) and three statements hardly could measure it. Connectiveness was measured using the scale for the measurement of social connecting.

5. There is high overuse of terms. Looks that connectiveness is equal to social connecting, and represents social affordance. While social affordance was used as synonymous with connective affordances. The immersive experience is mostly used for augmented reality or virtual reality technologies (Kuhail, M. A., ElSayary, A., Farooq, S., & Alghamdi, A. (2022). Exploring Immersive Learning Experiences: A Survey. Informatics 9(4), 75) adapted in online stores. Meantime, gamification is mostly related to engagement (De Canio, F., Fuentes-Blasco, M. and Martinelli, E. (2021). Engaging shoppers through mobile apps: the role of gamification, International Journal of Retail & Distribution Management, 49 (7), 919-940).  Finally, novelty seeking is a characteristic of participants, while characteristics of gamification could be a novelty as it was measured by the original scale (McLean, G., & Wilson, A. (2019). Shopping in the digital world: Examining customer engagement through augmented reality mobile applications. Computers in Human Behavior, 101, 210-224.).

6. Data analysis does not include means and standard deviations for variables. Probably they would add some light on the evaluation of immersive experience, purchase intention, and other variables. Some statistics like Cronbach’s alpha and Dillon–Goldstein’s ρ, etc., were not presented in the article. Plus, some model fit indicators could be presented (see Dash, G., & Paul, J. (2021). CB-SEM vs PLS-SEM methods for research in social sciences and technology forecasting. Technological Forecasting and Social Change, 173, 121092.)

Round 2

Reviewer 2 Report

Dear authors,

I’m thankful for your exhaustive comments about my notices. They clarify a lot to me, but readers will lack these arguments, since most of them were presented just to me, but were not included in the article.

I like the idea about modification of the title, but currently, I see the article with the old title. Thus, looks the change of title was dedicated only to me.

The modified title “Exploring the gamification affordances in online shopping with the heterogeneity examination through REBUS-PLS” does not put attention to gamification, but concentrates on gamification affordances. Such change excludes many gamification-related comments. However, another question appears – why do you have to write so intensively about gamification in section 2.1? Probably it would be enough to use information about gamification affordance, presented in 2.2.1. Moreover omitting 2.1. would enable you to present more information about variables, the model, scales, etc., and add the information that you presented to me personally due to the length of the article.

Thank you for adding the necessary information about model fit indicators - the goodness of fit (GoF) for the global model (which was not included previously). However, the current text rise more questions and doubts. You wrote “The goodness of fit (GoF) value of the global model is 0.76” on page 11 lines 499-500. However, this text contradicts the later text “For class 1, the GoF and R2 (purchase intension) values are 0.575 and 0.493 respectively, which are greater than those of the global model.” page 13 lines 568-569. The same happens for Class 2 lines 578-579, while GoF did not present for Class 3 at all. Finally, GoF for Class 2 is lower than the threshold of 0.7 (“Acceptable “good” values within the PLS-PM community are GoF >0.7” – taken from your comments to me).

Probably you should pay more attention to quite low values of SD in all statements. Probably that could indicate a low variance in the answers – most of the statements were evaluated using just two answers – 3 and 4. Such a situation could happen when respondents don’t know what to say due to unclear statements or the analyzed situation and select a neutral statement or a slightly positive one.
